# Oligonucleotide Fluorescence In Situ Hybridization: An Efficient Chromosome Painting Method in Plants

**DOI:** 10.3390/plants12152816

**Published:** 2023-07-29

**Authors:** Arrashid Harun, Hui Liu, Shipeng Song, Sumeera Asghar, Xiaopeng Wen, Zhongming Fang, Chunli Chen

**Affiliations:** 1Key Laboratory of Plant Resource Conservation and Germplasm Innovation in Mountainous Region (Ministry of Education), Institute of Rice Industry Technology Research, College of Agricultural Sciences, Guizhou University, Guiyang 550025, China; harun@gzu.edu.cn; 2Key Laboratory of Plant Resource Conservation and Germplasm Innovation in Mountainous Region (Ministry of Education), Institute of Agro-Bioengineering, College of Life Science, Guizhou University, Guiyang 550025, China; sumeraroy211@gmail.com (S.A.); wenxp@gzu.edu.cn (X.W.); 3National Key Laboratory for Germplasm Innovation and Utilization for Fruit and Vegetable Horticultural Crops, Hubei Hongshan Laboratory, Wuhan 430070, China; liuhui.dj@webmail.hzau.edu.cn (H.L.); crush77@webmail.hzau.edu.cn (S.S.)

**Keywords:** oligo-FISH, single-copy, repetitive sequence, probe labeling, oligonucleotide synthesis, traditional FISH, oligo-FISH applications, molecular cytogenetics

## Abstract

Fluorescence in situ hybridization (FISH) is an indispensable technique for studying chromosomes in plants. However, traditional FISH methods, such as BAC, rDNA, tandem repeats, and distributed repetitive sequence probe-based FISH, have certain limitations, including difficulties in probe synthesis, low sensitivity, cross-hybridization, and limited resolution. In contrast, oligo-based FISH represents a more efficient method for chromosomal studies in plants. Oligo probes are computationally designed and synthesized for any plant species with a sequenced genome and are suitable for single and repetitive DNA sequences, entire chromosomes, or chromosomal segments. Furthermore, oligo probes used in the FISH experiment provide high specificity, resolution, and multiplexing. Moreover, oligo probes made from one species are applicable for studying other genetically and taxonomically related species whose genome has not been sequenced yet, facilitating molecular cytogenetic studies of non-model plants. However, there are some limitations of oligo probes that should be considered, such as requiring prior knowledge of the probe design process and FISH signal issues with shorter probes of background noises during oligo-FISH experiments. This review comprehensively discusses de novo oligo probe synthesis with more focus on single-copy DNA sequences, preparation, improvement, and factors that affect oligo-FISH efficiency. Furthermore, this review highlights recent applications of oligo-FISH in a wide range of plant chromosomal studies.

## 1. Introduction

Fluorescence in situ hybridization (FISH) is a crucial technique in plant molecular cytogenetic research that allows the visualization of DNA or RNA sequence location on real chromosomes [1,2,3]. However, in situ hybridization (ISH) experiments using RNA labeled with radioactive tritium were first reported in 1969 [4] and the FISH method was first reported in the early 1980s [5]. FISH employs fluorophore-labeled DNA or RNA molecules as probes to produce a hybrid double-stranded molecule with signal in the genome that could be detected under a fluorescence microscope using a CCD (charge-coupled device) [6]. The success of FISH experiments largely depends on the probes used in the FISH experiment. While traditional cloned DNA probes, such as bacterial artificial chromosome (BAC), ribosomal DNA (rDNA), tandem repeats, and distributed repetitive sequences probes, have been used to construct karyotypes, identify chromosomes, and determine interspecies homoeologous relationships among various plant species [2,7,8,9,10], the reliability of traditional repetitive sequences in species with high percentages of repetitive sequences has been called into question due to the variability in available probes for a given species, as well as the complexity of preparing ordered BAC contigs covering the entire genome. Additionally, some species exhibit transposable elements within their repetitive sequences, further complicating the matter. As such, there is a need to reevaluate the use of traditional repetitive sequences in these cases. Recent advancements in next-generation sequencing (NGS) and DNA sequence synthesis technologies have led to the development of oligonucleotide fluorescence in situ hybridization (oligo-FISH), which uses synthetic oligonucleotides as probes [11,12,13].

Oligo probe synthesis is an important step of the oligo-FISH technique, which involves computational design yielding probe synthesis with fewer resources, saving time and improving the efficacy of chromosomal studies. Little work has been reported on the computational tools required to design the oligo probes, despite several advancements in these probes, and probe sets have been seen. However, previously, various tools were established for this purpose, including padlock probes [14], ribosomal RNA [15], and oligo pairs, but due to off-targeted and cross-hybridization effects, an incomplete sequential database, and limited experimental validation results, it is inevitable to combine experimental validations and computational predictions. The latest tools like ProbeDealer (MATLAB-based) [16], Chorus2 [17], and iFISH (Python-based) [18] undoubtedly reduce the required hyperparameters and configurations, but they still need advanced bioinformatic expertise. Therefore, it is recommended to merge the available computational predictions and experimental validations to overcome the above-mentioned constraints.

This review has extensively studied the major accomplishments in oligo probes. The recent applications are highlighted and well structured in the manuscript as oligo probe synthesis, preparation, and applications of oligo probes including chromosomal identification, karyotyping, and chromosomal rearrangements, with chromosomal translocations and fusion. The limitations and future perspectives are also highlighted.

## 2. De Novo Oligo Probe Synthesis

### 2.1. Brief Introduction of Oligo Probes

Oligonucleotides, which are small molecules composed of 20–50 nucleotides, have gained attention in the field of FISH experiments [1,19,20]. Currently, there are two types of oligo probes in plants: oligo probes based on single-copy sequences and oligo probes designed from repetitive DNA sequences. These probes consist of single-copy oligos or shorter sequences with repeated di-, tri-, or tetra-nucleotides, such as (AG)12 or (AGG)5, and are designed to target specific chromosomal segments or entire chromosomes. They can be used as a probe library and are easily synthesized, amplified, and labeled using PCR amplification, making them a cost-effective option. Shorter probes offer higher sensitivity, specificity, and ease of hybridization. Multiplex oligo probe designs are also available and are inspired by simple oligo probe design processes. The quality of the synthesis process depends on the synthesizer and the reagents used.

### 2.2. Comparison of Two Types of Oligo Probes

In plants, two types of oligo probes have their own advantages and disadvantages. Typically, single-copy oligo-based probes are used, which are synthesized from single-copy DNA sequences using a bioinformatics platform. These oligo sequences are labeled with fluorochrome, biotin-dUTP, or digoxigenin-dUTP through PCR amplification, generating oligo probes that have been used in FISH experiments [21]. Single-copy oligo probes provide flexibility for use on specific chromosome fragments or the entire chromosome as a library. It is easy to use for amplification via PCR for labeling, which reduces the cost of probe reuse. Single-copy oligo probes are valuable for studying chromosomal evolution, meiotic pairing, and recombination, which can distinguish homologous chromosomes and even homoeologous chromosomes from different cultivars [11]. However, the successful design of oligo probes needs to be achieved in plants that have completed genome sequencing. A lack of genome sequencing information hinders the application of single-copy oligo-based probes. 

Unlike oligo probes based on single-copy sequences, designing probes from repetitive DNA sequences does not require genome sequencing information [1]. These oligo sequences consist of microsatellite repeats, satellite repeats, and tandem repeats, and can be end-labeled with biotin-dUTP or digoxigenin-dUTP [1,12]. Furthermore, repetitive DNA oligo probes are commonly used in FISH to produce unique signals for individual chromosomes. This technique is useful for karyotyping and phylogenetic analysis, as well as for non-denaturing FISH (ND-FISH) [22]. Although repetitive DNA oligo probes are easier to prepare, single-copy shorter probes are more specific and have higher sensitivity. Oligo probe design has multiple well-developed applications, making the design process easy.

### 2.3. The Procedure and Rules of Oligo Probe Preparations

De novo oligo probe synthesis is a combined approach that starts with a computational search of single-copy potential oligos in the genome, and then sequential steps are performed. A method for de novo single-copy oligo probe synthesis is described in Figure 1 and Figure 2, which represent combined approaches of computational and experimental techniques. In brief, the process involves selecting potential single-copy oligos, synthesizing them, and labeling them with fluorescent markers [7,23]. However, first, single-copy oligos are computationally searched in the target genome, repetitive sequences are filtered out, and potential bulked oligos are selected using the Chorus2 platform (Figure 1). Next, the selected potential DNA sequences are bound with a specific pair of primers to their 3’ and 5’ ends for PCR amplification for transcription or reverse transcription for making a double strand, and then fluorescent markers are added (Biotin, digoxigenin or fluorescence probe) in wet-lab experiments [24] (Figure 2). Obtaining genomic data for oligo selection is possible through the genomic database, and the process has been greatly facilitated using next-generation sequencing (NGS) technology [1]. Depending on the purpose of the experiment, single-copy oligo sequences can be made into ‘oligo pool’ probes specific to any region of the chromosome [12,20]. The specificity of the probe is crucial for a single-copy oligo probe to determine the quality of the FISH experiment with specific FISH signals [25]. Therefore, selecting a well-assembled reference genome and suitable software or a suitable platform is essential to design an efficient single-copy oligo probe. It is important to follow other rules as well, such as maintaining a minimum space of typically 10 bases between probes to minimize overlapping between adjacent probes, maximize PCR quality, and minimize the mismatch ratio between the probe and target sequence [20]. For more information, Liu and Zhang (2021) [26] published a review article summarizing the various attributes of single-copy oligonucleotides and discussing comprehensive probe design platforms. The preparation process of oligo probes based on repetitive sequences is similar to that for single-copy sequences, with the main distinction being the filtering process of target sequences.

### 2.4. Key Aspects (Characteristics) That Influence Single-Copy Oligo Probe Performance

In order to achieve a successful oligo-FISH experiment, it is crucial to design high-quality single-copy oligo probes. The process involves minimizing repetitive sequences in the target sequences [2,27,28,29] and taking into account factors such as probe density, length, temperature, and quality of the chromosome preparation, which are applicable to all types of cells [11,12,30]. Previous research has shown that successful oligo-FISH in plants has been achieved with varying densities of oligo probes, ranging from 0.1 to 0.5 oligo/kilobases [1]. The number of oligos in a probe pool is also important, as higher-density probes generate brighter signals but at a higher cost. However, repeat detection and filtering are essential for improving single-copy oligo probes, especially in large and complex plant genomes [17,31,32]. To ensure consistent separation of the two FISH signals, a distance of 5–10 Mb is necessary between the two spots, with each spot containing a minimum of ~1000 oligos to ensure a strong and punctuated signal of a smaller chromosomal region [1]. Earlier studies on various plant species have indicated that a density of 0.1–0.5 oligos/kb is the most effective way to obtain a high-quality signal on condensed metaphase chromosomes. However, to achieve a strong signal, a higher density of approximately 2 oligos/kb is required for pachytene chromosomes, which are known to be extended 10–20 times more than metaphase chromosomes [1]. This was demonstrated in rice (*Oryza sativa*) where a probe with a density of 2 oligos/kb produced an excellent signal on pachytene chromosomes for specific chromosome identification [33], and cucumber (*Cucumis sativus*) produced stronger signals with 7.3 oligos/kb than 3.2 oligos/kb for megabase-size chromosome painting [12].

Both single and repetitive sequences oligo probes can be designed for plant species with sequenced genomes using various platforms such as Chorus2, iFISH, and Oligopaint [17,18,34]. These techniques offer several potential advantages in plant cytogenetic research, including the ability to design probes for plants without reference genomes. Researchers can use either chromosome-level or single-copy sequences in the target plant species or other genetically related plants to design oligo probes [11,12,30,35]. Oligo-FISH can also be used to design chromosome-level probes for detecting polyploid plant species, which demonstrate multiple-genome hybridization with varying levels of homology. The cross-hybridization of homologous or homoeologous chromosomes during probe design can be accounted for by designing each oligo from homoeologous chromosomes with a similar level of sequence similarity. This allows for a similar level of signal to be produced from each oligo, which can be useful in revealing the polyploid genomes [35]. However, oligo-FISH is a useful technique that can be applied to a wide range of plant species which provide valuable insights into their genome structure and organization, including polyploid species such as corn (*Zea perennis*) [11] and *Saccharum* [36,37].

## 3. Applications of Oligo-FISH in Plants 

Oligo-FISH has become a widely used technique in various molecular cytogenetic applications in plant species in recent years. Like traditional FISH, chromosome identification through chromosome painting is the key application of oligo-FISH, which is more powerful than traditional probes. However, chromosome painting is one of the most successful applications of this technique, and it has been reported in many plant species. Other applications of oligo-FISH in various plants have been systematically summarized in Table 1, Table 2 and Table 3. These include chromosome identification, karyotyping, and the determination of chromosome rearrangements, which allow for the construction of detailed chromosomal maps of plant species. These maps are important tools for understanding genome structural organization and evolution analysis in plants. Figure 3 illustrates the versatility of this technique and its potential to contribute to a wide range of research areas in plant molecular cytogenetics. The continued development and refinement of oligo-FISH techniques are likely to play increasingly important roles in advancing our understanding of plants’ chromosomal identification and their evolutionary history.

### 3.1. Chromosome Identification with Chromosomal Arms, Segments, and Centromere Sequence Markers

The oligo-FISH technique is a highly effective method for identifying and mapping chromosomes in various plant species, including those without reference genomes. It involves the use of specific single-copy DNA sequences for probes to identify whole chromosomes, chromosomal segments, and the long and short arms of chromosomes. This technique has been successfully applied in various plant species, such as sweet orange (*Citrus sinensis*) [77], calamondin (*C. microcarpa*) [78], cucumber (*Cucumis sativus*) [12], maize (*Zea mays*) [46], sorghum (*Sorghum bicolor*) [52], sugarcane (*Tripidium arundinaceum*) [52], wheat (*Triticum boeoticum*) [44], banana (Musa spp.) [56], potato (*Solanum tuberosum*) [35,48], tomato (*Solanum lycopersicum*) [35], and barley (*Hordeum vulgare*) [50]. The use of specific oligo probes has been allowed for the separation of Ab- and A-genome chromosomes in wheat (*Triticum boeoticum*) [44]. In addition, oligo-FISH has been used to identify all ten *Erianthus rufipilus* (*Saccharum* complex) centromeres, using satellite CEN137 monomers as probes [79]. Overall, the oligo-FISH technique is a powerful tool for chromosome identification with chromosome painting using any markers as probes, applicable for both model and non-model plants.

### 3.2. Karyotyping and Evolution 

Oligo-FISH is an important molecular cytogenetic technique that not only enables the identification of chromosomes via painting but also gives valuable insights for the karyotyping of plant species [80] and evolution of plant chromosomes across different species with repetitive and single-copy oligo sequences, respectively [1]. While comparative genetic linkage mapping has traditionally been used to study the syntenic and evolutionary relationships of homoeologous chromosomes among species [81], repetitive oligo probes have emerged as a more effective, flexible, and easy-to-use tool for such analyses. Many plant species have already successfully adopted this technique for karyotyping and identifying evolutionary relationships among them [11,38,39], as shown in Table 1. For example, Zhang et al. (2023) [82] recently used oligo-FISH to reveal the evolutionary effects of stable meiotic pairing behavior in different clones of cultivated sugarcane (*Saccharum spontaneum*). He et al. (2020) [63] revealed the extraordinarily conserved chromosomal synteny of *Citrus* species via oligo-FISH. Furthermore, oligo-FISH played an important role in the construction of ancestral chromosome karyotypes in *Cucumis* [83]. Overall, oligo-FISH is a valuable tool for studying the evolutionary relationships among related plant species with specific single-copy DNA sequences.

### 3.3. Chromosomal Rearrangement through Chromosomal Translocations and Fusion

Oligo-FISH mapping is a useful technique for identifying chromosomal translocations in various plant species with low-copy oligo probes. For example, it has been used to identify translocations between chromosome 9 and chromosome 11 in the indica rice (*Oryza sativa*) variety Zhongxian 3037 [33], as well as to measure the breakpoints of the 5B and 7B chromosomes during translocation in the wheat (*Triticum aestivum*) variety CM62 [41]. The oligo-GISH technique has also been used with a specific DNA probe to identify chromosome transmission in BC4 progenies during intergeneric hybrids between sugarcane (*Saccharum* spp.) and the *saccharum* complex (*Erianthus arundinaceus*) (Retz.) Jeswiet [59]. Recent research on oligo-FISH using a subgenome-specific interspersed repeat (IR) oligo probe revealed that four chromosomal translocations occurred between the A and B subgenomes during peanut (*Arachis hypogaea*) variety FS2020-2-1 polyploidization [58]. Overall, oligo-FISH is an important technique for chromosomal painting, mapping, evolution analysis, and the detection of chromosomal translocations in plants with specific probes.

## 4. Discussion

Oligo-FISH has emerged as a promising alternative to traditional FISH in plant molecular cytogenetic research, overcoming many of the limitations of traditional FISH methods [12,35,40,45,46,65,84,85,86]. This technique has proven useful in chromosomal identification, karyotyping, and the detection of chromosomal rearrangements through chromosomal fusion or segment translocation. Oligo-FISH has also contributed significantly to genomic and evolution analyses in both diploid and polyploid plants, as well as in studying chromosome pairing during meiosis [12,13,37,48]. Recent advancements in next-generation sequencing technology and probe designing tools have enabled the use of oligo probes for single and repetitive sequences, chromosomal segments, and entire chromosomes with all types of genomes, expanding the range of applications of oligo-FISH in plants. In summary, the combination of modern sequencing technologies, oligo probe designing tools, and the oligo-FISH technique provide researchers with a powerful arrangement for studying the molecular cytogenetics and genetic mechanisms underlying plant chromosome identification, karyotyping, and evolution analysis for the development of plants.

### 4.1. Advantages of Oligo-FISH 

Oligo probes which utilize both single-copy and repetitive DNA sequences offer several advantages over traditional probes, including a higher probe density, lower probe cost, stronger probe availability, and greater flexibility in probe designing, targeting both the single-copy and repetitive-copy basis of experimental needs. Furthermore, oligo-FISH can visualize multiple targets within a cell, such as mRNA (RNA FISH discussed later), using “probe sets” as well as single targets, using target-specific oligo probes [87].

### 4.2. Efficient Use of Oligo-FISH

In order to improve the efficiency and experimental results of oligo-FISH, it is necessary to pay attention to several aspects in the design of oligo probes. Firstly, different types of oligo probes need to be designed according to the purpose of the experiment. For example, the probes designed for the chromosomal segment may not be suitable for identifying chromosomal variations among plants, which require numerous oligos covering all individual chromosomes. Meanwhile, when designing probes and preparing chromosomes for oligo-FISH, it is crucial to also take into consideration the thermodynamic properties of the sample DNA, which can vary depending on the plant cells. Unfavorable conditions may result in failed probe and target hybridization, ultimately affecting the success of the oligo-FISH experiment [88]. Furthermore, plants with large genomes and big chromosome numbers often have a high proportion of repetitive sequences, making it challenging to obtain efficient single-copy oligo probes with low background noises. To address this challenge, a combination of sequence alignment and k-mer-based analysis can be applied during probe design to identify single-copy oligos. Repetitive sequences may not be effectively eliminated using sequence alignment alone [17]. To address this issue, the genomic shotgun sequencing method can be used to filter out repetitive sequences from the low-quality sequencing data of these genomes. The above approaches will be helpful to improve the efficiency of oligo-FISH for plants.

### 4.3. Limitation of Oligo-FISH

Although we can design oligo probes of both repetitive and single sequences for target species with the help of NGS and probe designing software, many plant species still face obstacles in adopting oligo-FISH for chromosomal analysis due to their big and diverse chromosome numbers and lack of genomic information, despite the potential benefits of this technique for improving crops cytogenetically [1]. For example, tropical fruit crops such as mango (*Mangifera indica*), pitaya (*Hylocereus undatus*), and cactus (*Selenicerus grandiflorus*) species currently lack the necessary resources for oligo probe design for chromosomal analyses with oligo-FISH. However, there is hope for the future, as the oligo-FISH technique has already been utilized for more complex and polyploid genomes such as wheat (*Triticum aestivum*) [39], sugarcane (*Saccharum officinarum*) [37,88], and potato (*Solanum tuberosum*) [35,48]. With advancements in next-generation sequencing technology and oligo probe design software, it is hoped that more crops will be able to take advantage of genomic and cytogenetic studies in the future.

## 5. Conclusions and Future Perspectives

FISH has been considered to be the most fundamental technique in plant cytogenetic research since 1969. However, the lack of robust DNA probes in many plant species has been a major challenge. With the development of next-generation sequencing technologies and bioinformatics tools, the design and synthesis processes of oligo probes for oligo-FISH have greatly improved, resulting in a more flexible and target-specific oligo probe designing, which has led to the achievement of target site determination in the genome. This has expanded its applications in the molecular cytogenetic field in plants, particularly in the areas of molecular identification of chromosomal variations, chromosomal rearrangement, and evolution detection. Moreover, oligo probes have alleviated the previous dilemma of the lack of universal DNA probes for many genetically related plant species, which was a limitation of traditional probes, making the oligo probe an important tool in plant molecular cytogenetic research. It also provides a good understanding of plants’ genomic structures, opening up new avenues for exploring the genetic mechanisms underlying plants’ development and evolution [1]. However, several advanced FISH techniques have gained popularity in the field of biology, significantly impacting research in molecular cytogenetics.

π-FISH is a highly efficient and robust fluorescence in situ hybridization (FISH) method called the π-FISH rainbow for detecting diverse biomolecules, including DNA, RNA, proteins, and neurotransmitters, with high sensitivity and specificity. The π-FISH rainbow method involves the use of target probes containing 2–4 complementary base pairs in the middle region, followed by secondary and tertiary U-shaped amplification probes, and a fluorescence signal probe. The versatility of the π-FISH rainbow was demonstrated by applying it to diverse species in frozen, paraffin, and whole-mount samples, and by combining it with other imaging technologies, such as vascular labeling. Additionally, the π-FISH rainbow was utilized to determine the spatial landscape of cells in intact tissue and to detect small genomic indels and breakpoints. The π-FISH rainbow was combined with the hybridization chain reaction (HCR) to detect short nucleic acid fragments, such as microRNA and the prostate cancer anti-androgen therapy-resistant marker ARV7 splicing variant [89].

Three-dimensional (3D) FISH has been used to construct the 3D structure of plant interphase nuclei, meiotic nuclei, and the nuclear disposition of hybrids [90,91]. With the increasing availability of plant genome sequencing and the design flexibility of oligo painting probes, 3D-FISH using oligo probes will become more diverse in studying the three-dimensional spatial structure of plant chromosomes and chromatin with immunolabeling and high-resolution microscopy approaches in the future. 

RNA-FISH is another type of oligo-FISH that uses oligonucleotide probes to detect mRNAs with single-molecule resolution and sRNA [92]. By combining single-copy oligonucleotide probes, RNA FISH can be used to detect any kind of RNA, including tRNA. RNA imaging in live cells and 3D combined with immunofluorescence may become a new direction of plant cytogenetic research, providing researchers with a powerful tool to study the spatial and temporal regulation of gene expression in plants. This will expand our understanding of plant gene regulation and development at the molecular level, opening up new avenues for exploring the genetic mechanisms underlying plant growth and adaptation in changing environments. 

Live imaging is a novel FISH technique called LiveFISH, which uses oligo probes to target genomic sequences in living cells, and has been developed for imaging telomeric sequences in tobacco (*Nicotiana benthamiana*) [93,94,95]. By combining CRISPR-mediated LiveFISH with fusion fluorescent protein transgenic approaches, it will be possible to trace defined DNA sequences and proteins in living plant cells simultaneously. This has potential applications to study the dynamics of chromatin during special cell activities such as cell fusion, mitosis, and meiosis. However, oligo-FISH based on chromosome and oligonucleotide synthetic probes remains the cornerstone of many types of FISH. Combined with the above diversified FISH technologies, oligo-FISH will be widely used in plants in the future.

## Figures and Tables

**Figure 1 plants-12-02816-f001:**
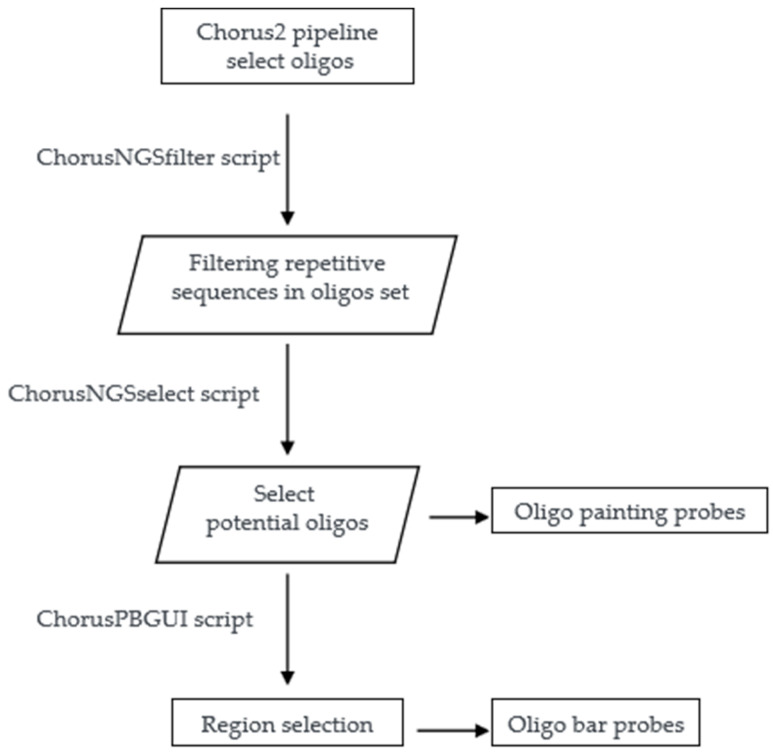
The oligo search, filtering, and selecting of potential oligos in the genome using the Chorus2 pipeline.

**Figure 2 plants-12-02816-f002:**
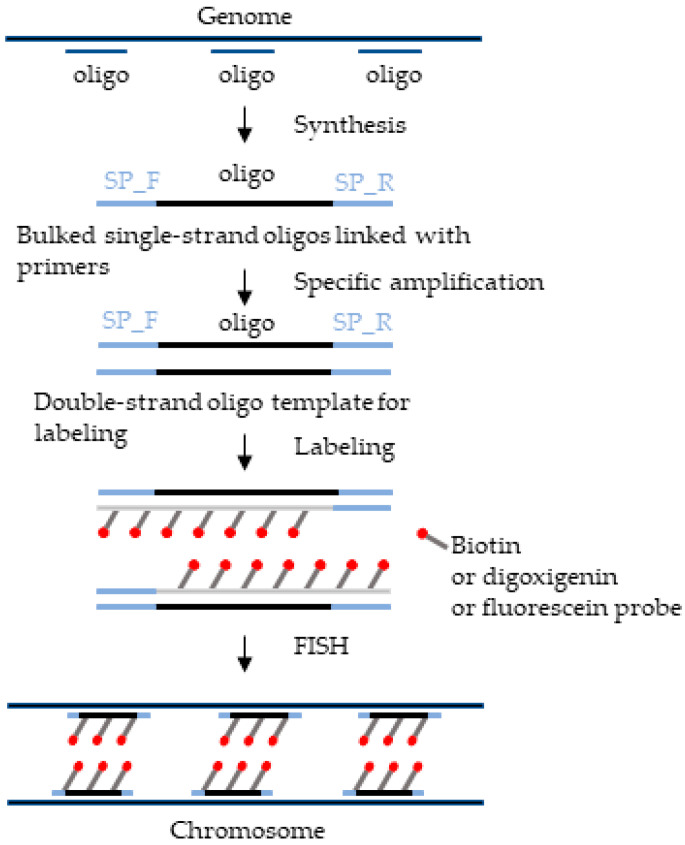
The schematic diagram represents oligo probe library enrichment, PCR amplification for primer binding, fluorophore labeling for de novo probe synthesis, and FISH experiment being performed with developed oligo probes.

**Figure 3 plants-12-02816-f003:**
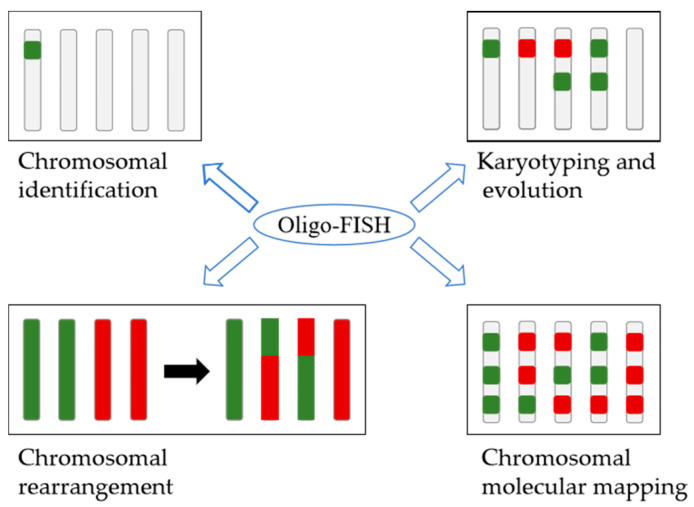
Applications of oligo-FISH in plants.

**Table 1 plants-12-02816-t001:** The applications of both oligo-FISH and traditional FISH techniques on economical plants are summarized.

Serial No.	Species	FISH-Probe	Application	Signal Effect	Cell	Ref.
1.	Rice (*Oryza sativa*)	Two specific oligo and 5S-rDNA-based probes	Karyotyping, detection of chromosomal variations, and chromosome translocation, mainly transposition	Strong	Metaphase and pachytene	[38]
		Chromosome-9-specific oligo pools, CentO repetitive, and 45S-rDNA-based probes	Chromosome identification and translocation detection	Strong	Interphase, prometaphase, premeiotic interphase, zygotene, pachytene, and microspore	[33]
2.	Wheat (*Triticum aestivum*)	Specific oligo probes, named pTa-535, oligo-18, pTa-275, pSc119.2, SC5A-479, and SC5A-527, derived from both single-copy and tandem repeats	Karyotyping, detection of meiotic recombination, and structural alterations	Mixed	Metaphase	[39]
		Synthesized oligo probe library contains 27,392 oligo pools	Chromosome identification and detection of chromosomal rearrangements	Mixed	Metaphase	[40]
		Twenty specific oligo probes with lengths of 20–60bp	Chromosome identification, karyotyping, and chromosomal translocation detection	Mixed	Metaphase	[41]
		One hundred and twenty specific oligo probes based on new tandem repeats	Chromosomal mapping and distinguishing A-, B-, and D-genome chromosomes	Mixed	Metaphase	[42]
		Repetitive oligo probes	Chromosomal mapping and distinguishing chromosomes of wheat (*T. aestivum*)-*D. villosum* amphiploid	Mixed	Metaphase	[43]
3.	Wheat (*Triticum boeoticum*)	Specific oligo probes, named Oligo-pTa535-HM, Oligo-pSc119.2-HM, (ACT)7, (CTT)7, (GAA)7, Oligo-pTa713,,(CAG)7, and (CAC)7	Chromosome identification and comparison between Ab- and A-genome chromosomes	Mixed	Metaphase	[44]
4.	Wheat (*Triticum aestivum* L.)-*Th. Ponticum, Th. intermedium* partial amphiploid lines	Two specific oligo probes, named oligo-B11 and oligo-pThp3.93	Chromosome mapping, discrimination of chromosomes of *Th. elongatum*, *Th. intermedium*, and *Th. ponticum* in wheat backgrounds and chromosomal translocation detection	Mixed	Metaphase	[45]
5.	Maize (*Zea mays*)	Two oligo bar codes	Chromosome identification, karyotyping, and chromosomal translocation detection	Mixed	Metaphase	[11]
		Chromosome-1-to-10-specific unique-sequence-based oligo pools and repetitive-sequence-based probe named CentC	Chromosome identification and rearrangement analyses	Mixed	Metaphase, pachytene, and interphase	[46]
		Oligo probes specific to chromosome 10 haplotype	Chromosome painting and visualization of COs (cross overs)	Mixed	Metaphase	[13]
		Microsatellite, subtelomeric, 5S rRNA, Cent4, CentC, knob, NOR, pMTY9ER-telomere-associated sequence, and tandem-repeated DNA sequence 1(TR-1)-based repetitive probes	Chromosome identification, mapping, karyotyping, chromosomal structure, and behavior analyses	Mixed	Pachytene, late prophase I, and metaphase I	[47]
6.	Potato (*Solanum tuberosum*)	Single-copy DNA sequence-based oligo probes	Identification of individual chromosomes, both diploid and polyploid of potato, homologous chromosomes of tomato (*S. lycopersicum*) and eggplant (*S. melongena*), karyotyping and translocation detection	Mixed	Metaphase	[35]
		Four chromosome-specific oligo probes	Karyotyping, meiotic pairing, and translocation detection	Mixed	Metaphase, early metaphase, early leptotene, late leptotene, zygotene, pachytene, diplotene, and diakinesis	[48]
7.	Rye (*Secale cereale*)	Oligo probes, named Oligo-1162, pSc200, and pSc250	Detection of rye chromosomes from wheat (*Triticum aestivum*) background	Strong	Metaphase	[49]
8.	Wheat×rye hybrid (*Triticum aestivum* × *Secale cereale*)	Nine oligo probes with specific repeat sequences	Identification of individual chromosomes and karyotyping	Strong	Metaphase	[23]
9.	Barely (*Hordeum vulgare*)	Seven chromosome-specific oligo pools	Chromosomal mapping of wheat and barley, karyotyping of other Triticeae, homologous and non-homologous chromosomal rearrangements in Triticeae, and wheat–barley synteny identification	Strong	Metaphase	[25]
		BAC and 5S rDNA probes	Identification of individual chromosomes and karyotyping	Strong	Metaphase and pachytene	[50]
10.	Brassica (*Brassica rapa*)	Three satellite repeat sequences from radish (*R. sativus*)-based probes	Chromosomal mapping	Weak	Metaphase	[51]
11.	Radish (*Raphanus sativus*)	Three satellite repeat sequences, and 45S- and 5S-rDNA-based probes	Karyotyping	Mixed	Metaphase	[51]
12.	Cucumis (*Cucumis sativus*)	Chromosome-1-and-4-specific oligo probes	Chromosome identification, homoeologous relationship detection among cucumber, *C. hystrix*, and *C. melo* chromosomes	Strong	Metaphase and pachytene	[24]
		Chromosomal segments and arm-specific oligo pools	Chromosome identification, chromosomal pairing, rearrangement, and evolution analyses	Strong	Metaphase, zygotene, and pachytene	[12]
13.	Sugarcane (*Tripidium arundinaceum*)	Maize chromosome painting probes (MCPs), 5S rDNA, and 35S rDNA	Identification of chromosome and karyotyping	Mixed	Metaphase	[52]
14	Sugarcane (*Saccharum officinarum*)	Chromosome-1-to-10-specific oligo pools	Identification of chromosomes, novel cytotypes, and chromosomal rearrangement analyses	Mixed	Metaphase	[53]
		Chromosome-specific oligo probes	Chromosomal rearrangements, karyotyping, and translocation detection	Strong	Metaphase	[37]
15.	Strawberry (*Fragaria vesca*)	Chromosome-specific bulked oligo probes, and 45S and 5S rDNA probes	Identification of chromosomes, chromosomal mapping, and karyotyping	Mixed	Metaphase	[54,55]
16.	Banana (Musa spp.)	Nineteen chromosomes/chromosome-arm-specific oligo pools and 45S rDNA probes	Identification of chromosomes, molecular karyotyping, and evolution analyses	Mixed	Metaphase and pachytene	[56]
17.	Beans (*Phaseolus vulgaris*, *Vigna angularis*, *Vigna unguiculata*)	Two oligo probes named Pv2 and Pv3, BAC and 35S rDNA probes	Identification of chromosomes, karyotyping, chromosomal rearrangements, evolution, cytogenetics maps, and syntonic relationship analyses	Mixed	Metaphase	[57]
18.	Peanut (*Arachis hypogaea*)	Subgenome-specific interspersed repeat (IR) oligo probes	Identification of chromosomes, genomic relationships, and chromosomal variant analyses	Strong	Metaphase	[58]
19.	Saccharum spp. × *Erianthus arundinaceus* hybrid	EaHN92 and HN92-105 (*E. arundinaceus*) genomic DNA sequence-based oligo probes	Identification of chromosomes and chromosome transmission detection	Strong	Metaphase	[59]
20.	Buckthorn (*Hippophaë rhamnoides*)	Oligo probes specific to (AG3T3)3 repetitive sequences, 5S rDNA, and (TTG)6 repetitive-sequence-based probes	Chromosome identification and karyotyping	Strong	Metaphase and anaphase	[60]
21.	Sour cherry (*Prunus* *Cerasus*)	Oligo probes from Arabidopsis telomere repetitive sequence, and centromere repetitive-sequence-based	Chromosomal mapping and karyotyping	Mixed	Metaphase	[61]
22.	Fabaceae (*Robinia pseudoacacia*, *R. pseudoacacia*, *R. pseudoacacia*, *Styphnolobium japonicum*, *Amorpha fruticose)*	Two specific oligo probes, repeat sequence (AG3T3)3, and 5S rDNA-based probes	Chromosome identification and karyotyping	Strong	Metaphase	[62]

**Table 2 plants-12-02816-t002:** The applications of both oligo-FISH and traditional FISH techniques on horticultural plants are summarized.

Serial No.	Species	FISH-Probe	Application	Signal Effect	Cell	Ref.
1.	Citrus (*Citrus maxima*)	Nine chromosome-specific oligo probes, repeats, and 45s- and 5S-rDNA-based probes	Identification of chromosomes, karyotyping, and chromosomal syntonic relationship analyses	Mixed	Metaphase	[63]
2.	Japanese morning glory (*Ipomoea nil*)	Four specific oligo probes, 45s- and 5S-rDNA-based probes	Identification of chromosomes, pseudochromosomes, karyotyping, chromosomal variation, and evolution analyses	Mixed	Metaphase	[64]
3.	Poplar (*Populus trichocarpa*)	Oligo probes specific to chromosome 19	Chromosome painting, mapping, and chromosomal pairing detection	Mixed	Metaphase and pachytene	[65]
		Complete set of 19 chromosome painting probes	Identification of chromosomes, karyotyping, chromosome pairing, collinearity, and evolution analyses	Mixed	Metaphase and pachytene	[66]
4.	Lupin (Lupinus)	Oligo probes based on chromosome-arm-specific and BAC probes	Karyotyping, chromosome evolution, and translocation detection	Mixed	Metaphase	[67]
5.	Chrysanthemum (*Chrysanthemum nankingense*, *C. lavandulifolium*, *C. dichrum*, *C. indicum* cv. Henan, *C. indicum* cv. Fujian, *C. indicum* cv. Hubei, *C. potentilloides*, and *C. rhombifolium*)	Specific oligo probes, and 5S- and 45S-rDNA-based probes	Chromosomal mapping and karyotyping	Strong	Interphase nuclei and metaphase	[68]

**Table 3 plants-12-02816-t003:** The applications of both oligo-FISH and traditional FISH techniques on flowering plants and grass species are summarized.

Serial No.	Species	FISH-Probe	Application	Signal Effect	Cell	Ref.
1.	Aegilops (*Aegilops* *umbellulata*, *Aegilops markgrafii*, *Aegilops comosa*, *Aegilops uniaristata*)	Oligo probes, named pSc119.2 and pTa71, in combination with (AAC)5, (ACT)7, and (CTT)12 repetitive sequences	Chromosomal variation and karyotyping	Mixed	Metaphase	[69]
2.	Wheatgrasses (*Thinopyrum intermedium*)	Specific oligo probes and 5S rDNA	Identification of chromosomes and karyotyping	Strong	Metaphase	[70]
3.	Siberian wild rye (*Elymus sibiricus*)	Two specific oligo probes, repeats, and rDNA probes	Identification of chromosomes, Karyotyping, and ideogram constructing	Mixed	Metaphase	[71]
4.	Wild sugarcane *(Saccharum spontaneum)*	Chromosome-specific oligo barcode	Chromosome distinguishing, karyotyping, and rearrangement analyses	Strong	Metaphase	[36]
		Seventeen oligo barcodes, sorghum-derived oligo probes, 45S- and 5S-rDNA-based probes	Chromosome identification, Karyotyping, and chromosomal rearrangement analyses	Strong	Metaphase	[72]
5.	Antarctic hairgrass (*Deschampsia antarctica* Desv.)	Repeated DNA, 45s- and 5S-rDNA-based probes	Identification of chromosomes and karyotyping	Strong	Metaphase	[73]
6.	Grass (*Roegneria ciliaris*)	Oligo multiplexing probes	Chromosome identification and identification of pan and core karyotyping	Strong	Metaphase	[74]
7.	Araliaceae	5S-, 45S-rDNA-, and telomeric-repeat-based probes	Identification of chromosomes and karyotyping	Strong	Metaphase	[75]
8.	Duckweeds (Lemnaceae)	Chromosome-specific oligo probes and BAC probes	Identification of chromosomes, chromosomal rearrangement, and evolution analyses	Mixed	Metaphase	[76]

## Data Availability

This review article does not contain any original data or analyses. All data used in this article were obtained from previously published sources, which are cited appropriately.

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
