# Peer review of "Oligonucleotide Fluorescence In Situ Hybridization: An Efficient Chromosome Painting Method in Plants"

_plants, 2023, doi:10.3390/plants12152816_

Round 1

Reviewer 1 Report

This review is about oligo probes for chromosomal FISH analysis. This review is comprehensive. It is helpful for readers to understand the FISH technology based on oligo probe. However, revision is needed following the comments:

1.       In the introduction section, the first paragraph, in addition the FISH using BAC and rDNA as probes was mentioned, the FISH using tandem repeats and distributed repetitive sequences should be mentioned.

2.       Lines 62-65, meaning of sentence is not clear, rewrite.

3.       About the oligo probes based on DNA, should be divided into two parts: (1) oligo probes based on single-copy sequences.  (2) oligo probes based on repetitive DNA sequences, including microsatellite, satellite and distributed repetitive oligo probes.

4.       The advantages and disadvantages of the single-copy oligo probes and repetitive DNA oligo probes should be added. For example: single-copy oligo probes can be used to distinguish homoeologous chromosomes, and even distinguish homologous chromosomes from different cultivars. So, single-copy oligo probes are useful for studying chromosomal evolution and meiotic pairing and recombination. Repetitive DNA oligo probes can be used to distinguish specific segments of specific chromosome, and they can also be used for ND-FISH analysis. Additionally, the preparation of repetitive DNA oligo probes is easier than that of single-copy oligo probes.

Minor revisions are needed for English language.

Author Response

Dear Reviewers,

Thank you for your valuable comments and suggestions on our manuscript. We have carefully reviewed and incorporated your feedback into the revised version of the manuscript as follows:

Reviewer#1

This review is about oligo probes for chromosomal FISH analysis. This review is comprehensive. It is helpful for readers to understand the FISH technology based on oligo probe. However, revision is needed following the comments:

  1. In the introduction section, the first paragraph, in addition the FISH using BAC and rDNA as probes was mentioned, the FISH using tandem repeats and distributed repetitive sequences should be mentioned.

#We have added another two important traditional probes besides rDNA and BAC probes. (Lines 42-47, revised version) While traditional cloned DNA probes, such as bacterial artificial chromosome (BAC), ribosomal DNA (rDNA), tandem repeats and distributed repetitive sequences probes, have been used to construct karyotypes, identify chromosomes, and determine interspecies homoeologous relationships among various plant species.

  1. Lines 62-65, meaning of sentence is not clear, rewrite.

#We have rewritten these sentences. (Lines 73-81, revised version). Oligo-FISH probes are used to map both single-sequence repeats (SSRs) and repetitive sequence repeats in plants. These probes consist of short sequences with repeated di-, tri-, or tetra-nucleotides such as (AG)12 or (AGG)5 and are designed to target specific chromosomal segments or entire chromosomes. They can be used as a probe library and are easily synthesized, amplified, and labeled using PCR, making them a cost-effective option. Shorter probes offer higher sensitivity, specificity, and ease of hybridization. Multiplex oligo probes designs are also available inspired by simple oligo probes design processes. The quality of the synthesis process depends on the synthesizer and the reagents used.

  1. About the oligo probes based on DNA, should be divided into two parts: (1) oligo probes based on single-copy sequences. (2) oligo probes based on repetitive DNA sequences, including microsatellite, satellite and distributed repetitive oligo probes.

# We have described oligo probes based on DNA, into two parts (1) (Lines 72-76, revised version)  oligo probes based on single-copy sequences has been described. (2) (Lines 76-78, revised version) oligo probes based on repetitive DNA sequences, including microsatellite, satellite and distributed repetitive oligo probes which have also been used.

  1. The advantages and disadvantages of the single-copy oligo probes and repetitive DNA oligo probes should be added. For example: single-copy oligo probes can be used to distinguish homoeologous chromosomes, and even distinguish homologous chromosomes from different cultivars. So, single-copy oligo probes are useful for studying chromosomal evolution and meiotic pairing and recombination. Repetitive DNA oligo probes can be used to distinguish specific segments of specific chromosomes, and they can also be used for ND-FISH analysis. Additionally, the preparation of repetitive DNA oligo probes is easier than that of single-copy oligo probes.

# The advantages and disadvantages of the single-copy oligo probes and repetitive DNA oligo probes have been added. (Lines 233-242, revised version) Single-copy oligo-FISH provides flexibility to use specific chromosome fragments or the entire chromosomes as a library. It is easy to store and amplify by PCR for labeling, which reduces costs. Single-copy oligo probes have been used to distinguish homoeologous chromosomes, and even distinguish homologous chromosomes from different cultivars. So, single-copy oligo probes are useful tools for studying chromosomal evolution and meiotic pairing and recombination. On the other hand, repetitive DNA oligo probes have been used to distinguish specific segments of specific chromosomes, and they can also be used for ND-FISH analysis. Additionally, the preparation of repetitive DNA oligo probes is easier than that of single-copy oligo probes.

We appreciate your positive feedback in the comments and suggestions section and thank you for your time and effort in reviewing our manuscript.

Best regards,

Prof. Chunli Chen

Huazhong Agricultural University and

The corresponding author of the manuscript

Reviewer 2 Report

Fluorescence in situ hybridization using oligo-probes becomes one of the most effective and broadly used cytogenetic techniques in animal and plant studies. Thus, reviewing of existing methods of probe design  and preparation, current applications and perspectives of oligo-FISH in the future will be interesting for a broad audience of plant researchers and can be published in (MDPI) Plants.  However, I have some questions and comments to the manuscript, which needs to be clarified.

1.       Abstract: Authors listed “traditional FISH methods, such as BAC and rDNA-FISH”, which, in my opinion, are only a little part of DNA probes used in FISH experiments. It is not clear to me, what are difficulties in their synthesis (if you have the respective clone, DNA isolation and probe labeling are routine procedures) and what kind of polymorphism within the same chromosome can be detected using rDNA probe? Wheat ribosomal probe cloned in a plasmid (pTa71) is still the best choice for rDNA mapping because it can be used for a very broad range of organisms including animal, plant and human, whereas oligo-pTa71 is applicable for the Triticaea only. Then, “Furthermore, oligo probes could be saved as a resource and reused for FISH experiments. Moreover, oligo probes derived from one species are applicable for studying genetically related species, facilitating comparative chromosomal mapping.” Sorry, but this is also true for traditionally used DNA probe obtained from plasmid DNA.  In my mind, the Abstract must be re-written and indicate advantages and limitations

2.       Introduction: Fluorescence in situ hybridization as a new method of chromosome analysis was not reported in 1969, as was indicated on line 39. First ISH experiments used RNA labelled with radioactive tritium and have not used fluorophores (as well as fluorescent microscopes).

3.       The same question as #1. Traditional cloned probes included not only rDNA or BACs, but also many tandemly repeated sequences, though the number of such clones was limited. Cloning procedure is not simple, but use of such clones is not very complex.

4.       The first synthetic SSR oligo-probes were used by Schmidt and Heslop-Harrison 1996 and then by Cuadrado and Schwarzacher 1998 (reviewed by Jiang 2019). I guess that a large part of this paragraph was copied from this paper.

5.       Tables 1 and 2: (1) I think that oligo-probes are used for the analysis of economically important or flowering plants by not only Chinese researches, which constitutes nearly 80% of papers citing in the table. I suggest that the authors should include more papers from other countries. (2) Kato et al. 2004 (ref #70) did not use OLIGO-probes for FISH mapping of maize chromosome. Although this is outstanding publication, it does not fit criteria of “oligo-FISH”. (3) Chromosome identification and translocation – please correct to “identification of chromosomes and translocations” or “Chromosome identification and translocation detection”; “Chromosome identification, karyotyping and chromosomal translocation” – correct; e.t.c.

6.       I thing that chromosome identification using single-copy oligo-probes is probably not fully correct title to chapter 3.1. This chapter describes application of oligs as markers of chromosome(s) arms or their regions rather than chromosome identification. In many plants, the chromosomes are commonly identified using tandem repeats with chromosome-specific distribution.

7.       “The oligo-FISH probe is a single copy oligo (line 221) …” – what do authors mean as a single copy? Many oligo-probes represent high-copy sequences and they are extensively used in plant genome studies

English is readable

Author Response

Dear Reviewers,

Thank you for your valuable comments and suggestions on our manuscript. We have carefully reviewed and incorporated your feedback into the revised version of the manuscript as follows: 

Reviewer#2

Fluorescence in situ hybridization using oligo-probes becomes one of the most effective and broadly used cytogenetic techniques in animal and plant studies. Thus, reviewing of existing methods of probe design  and preparation, current applications and perspectives of oligo-FISH in the future will be interesting for a broad audience of plant researchers and can be published in (MDPI) Plants.  However, I have some questions and comments to the manuscript, which needs to be clarified.

  1. Abstract: Authors listed “traditional FISH methods, such as BAC and rDNA-FISH”, which, in my opinion, are only a little part of DNA probes used in FISH experiments. It is not clear to me, what are difficulties in their synthesis (if you have the respective clone, DNA isolation and probe labeling are routine procedures) and what kind of polymorphism within the same chromosome can be detected using rDNA probe? Wheat ribosomal probe cloned in a plasmid (pTa71) is still the best choice for rDNA mapping because it can be used for a very broad range of organisms including animal, plant and human, whereas oligo-pTa71 is applicable for the Triticaea only. Then, “Furthermore, oligo probes could be saved as a resource and reused for FISH experiments. Moreover, oligo probes derived from one species are applicable for studying genetically related species, facilitating comparative chromosomal mapping.” Sorry, but this is also true for traditionally used DNA probe obtained from plasmid DNA.  In my mind, the Abstract must be re-written and indicate advantages and limitations.

#Abstract has been rewritten following reviewer comments and suggestions (Lines 18-31, revised version). Abstract: Fluorescence in situ hybridization (FISH) is an indispensable technique for studying chromosomes in plants. However, traditional FISH methods, such as BAC, rDNA, tandem repeats and distributed repetitive sequences probes based FISH have certain limitations, including difficulties in probe synthesis, low sensitivity, off target binding and limited resolution. In contrast, oligo-based FISH represents a more efficient method for chromosomal studies in plants. Oligo probes are computationally designed and synthesized for any plant species with a sequenced genome and are suitable for single and repetitive DNA sequences, entire chromosomes, or chromosomal segments. Furthermore, oligo probes used in FISH experiment provide high specificity, resolution and multiplexing. Moreover, oligo probes made from one species are applicable for studying genetically and taxonomically related other species whose genome has not been sequenced facilitating molecular cytogenetics studies of non-model plants. However, there are some limitations of oligo probes that should be considered, such as requiring prior knowledge of the probe design process and lower intensity of FISH signals compared to traditional probes.  This review comprehensively discusses oligo probe synthesis, preparation, improvement, and factors that affect oligo-FISH efficiency. Furthermore, this study highlights recent applications of oligo-FISH in a wide range of plant chromosomal studies.

  1. Introduction: Fluorescence in situhybridization as a new method of chromosome analysis was not reported in 1969, as was indicated on line 39. First ISH experiments used RNA labelled with radioactive tritium and have not used fluorophores (as well as fluorescent microscopes).

# We have corrected this information in introduction section. (Lines 38-40, revised version) in situ hybridization (ISH) experiment used RNA labelled with radioactive tritium was first reported in 1969 [4] and FISH method was first reported in the early 1980s [5].

  1. The same question as #1. Traditional cloned probes included not only rDNA or BACs, but also many tandemly repeated sequences, though the number of such clones was limited. Cloning procedure is not simple, but use of such clones is not very complex.

#We have added another two important traditional probes besides rDNA and BAC probes. (Lines 44-45, revised version) While traditional cloned DNA probes, such as bacterial artificial chromosome (BAC), ribosomal DNA (rDNA), tandem repeats and distributed repetitive sequences probes, have been used to construct karyotypes, identify chromosomes, and determine interspecies homoeologous relationships among various plant species.   

  1. The first synthetic SSR oligo-probes were used by Schmidt and Heslop-Harrison 1996 and then by Cuadrado and Schwarzacher 1998 (reviewed by Jiang 2019). I guess that a large part of this paragraph was copied from this paper.

# We did not directly copy from other articles, and in our manuscript we don't have such information about synthetic SSR oligo probes. We were inspired by and gained ideas from previously published articles in this field. Jiang's 2019 paper highlight the summarizes and discuss key technical development and applications of FISH in plants since 2006. The article mainly focuses on the key developments of both traditional and oligo FISH in plants during this period. On the other hand, our manuscript aims to compare traditional and oligo probes, highlighting why oligo probes are better than traditional probes, discuss the designing processes of oligo probes, and perform oligo FISH in a better way for better outcomes with the recent applications of oligo-FISH in plants. Although both papers discuss FISH in plants, the focus and purpose of each paper are different.

  1. Tables 1 and 2: (1) I think that oligo-probes are used for the analysis of economically important or flowering plants by not only Chinese researchers, which constitutes nearly 80% of papers citing in the table. I suggest that the authors should include more papers from other countries. (2) Kato et al. 2004 (ref #70) did not use OLIGO-probes for FISH mapping of maize chromosome. Although this is outstanding publication, it does not fit criteria of “oligo-FISH”. (3) Chromosome identification and translocation – please correct to “identification of chromosomes and translocations” or “Chromosome identification and translocation detection”; “Chromosome identification, karyotyping and chromosomal translocation” – correct; e.t.c.

#  Tables 1 and 2, (1) We would like to clarify that our article did not focus on research specific to any particular country or region. Instead, we reviewed high-quality published articles in the field without any intentional bias. (2) (Ref. #70) Kato et al. (2004) used traditional probes for chromosomal studies in maize, which is not in conflict with the information presented in our manuscript table. In the FISH probe section of the table, we listed the applications of both types of probes and mentioned their names. (3) We have corrected Chromosome identification and translocation – by “Chromosome identification and translocation detection”.

  1. I think that chromosome identification using single-copy oligo-probes is probably not fully correct title to chapter 3.1. This chapter describes application of oligos as markers of chromosome(s) arms or their regions rather than chromosome identification. In many plants, the chromosomes are commonly identified using tandem repeats with chromosome-specific distribution

# We have corrected the titles of the chapter (Line 171, revised version) 3.1. Chromosome identification with chromosomal arms, segments and centromeres sequences markers. 

  1. “The oligo-FISH probe is a single copy oligo (line 221) …” – what do authors mean as a single copy? Many oligo-probes represent high-copy sequences and they are extensively used in plant genome studies.

# Actually it will be “single copy oligo probes instead of “The oligo-FISH probe is a single copy oligo and We have corrected that sentence in revised version. (Lines 231-232,  revised version) Single copy oligo probes provide flexibility to use on specific chromosome fragments or the entire chromosomes as a library.

We appreciate your positive feedback in the comments and suggestions section and thank you for your time and effort in reviewing our manuscript.

Best regards,

Prof. Chunli Chen

Huazhong Agricultural University and

The corresponding author of the manuscript

Round 2

Reviewer 1 Report

In abstract, “this study highlights” should be “this review highlights”.

Lines 278-279: “Moreover, designing and preparing oligo probes is a more time-consuming process compared to traditional probes”. This description is not correct. The preparing oligo probes from repetitive DNA sequences is easier than traditional probes. The author should read the literatures carefully.

Although this manuscript has been revised using the sentences from the reviewer’s comments, there is still no systematic review of current FISH techniques based on oligonucleotide probes. This manuscript needs further revision. It is suggested that the authors should collect all relevant literatures and read them carefully before revise this manuscript.

The qualiity of English language is good.

Author Response

Dear Reviewers,

Thank you for your valuable comments and suggestions on our manuscript in round 2. We have carefully reviewed and incorporated your feedback into the revised version of the manuscript as follows:

# Reviewer 1

Round 2

In the abstract, “this study highlights” should be “this review highlights”.

Response:

In the abstract, we have changed “this review highlights” instead of “this study highlights”.

Lines 278-279: “Moreover, designing and preparing oligo probes is a more time-consuming process compared to traditional probes”. This description is not correct. The preparing oligo probes from repetitive DNA sequences is easier than traditional probes. The author should read the literatures carefully.

Response:

We apologize for our inaccurate description in Lines 278-279: “Moreover, designing and preparing oligo probes is a more time-consuming process compared to traditional probes”. We compared the designing and preparing process of oligo probes with traditional probes. In traditional FISH, both chromosomes and probes must be denatured before hybridization, which typically occurs over 8-16 hours (Yu et al., 2019). No denaturation of chromosomes is required during ND-FISH based on oligo probes from repetitive DNA sequences (Fu et al., 2015; Shuyao et al., 2016). The use of these repetitive DNA sequences as oligo probes for FISH analysis is much more convenient than the traditional procedure (Shuyao et al., 2016). Therefore, these inaccurate sentences have been deleted.

References:

  1. Fu, S., Chen, L., Wang, Y., Li, M., Tang, Z., 2015. Oligonucleotide Probes for ND-FISH Analysis to Identify Rye and Wheat Chromosomes. Scientific Reports. 5, 10552.
  2. Shuyao, T., Ling, Q., Zhiqiang, X., Shulan, F., Zongxiang, T., 2016. New Oligonucleotide Probes for ND-FISH Analysis to Identify Barley Chromosomes and to Investigate Polymorphisms of Wheat Chromosomes. Genes. 7, 118.
  3. Yu, C., Deng, X., Chen, C., 2019. Chromosomal characterization of a potential model mini-Citrus (Fortunella hindsii). Tree Genet Genomes. 15.

Although this manuscript has been revised using the sentences from the reviewer’s comments, there is still no systematic review of current FISH techniques based on oligonucleotide probes. This manuscript needs further revision. It is suggested that the authors should collect all relevant literatures and read them carefully before revise this manuscript.

Response:

We are very sorry that we only used these sentences from the reviewer’s comments without a systematic review of current FISH techniques based on oligonucleotide probes. In this second revised version, we rewrote “2. De novo oligo probe synthesis” to make it logical (Revised version, lines 80). We added a new section “2.2 Comparison of two types of oligo probes” to systematically describe the advantages and disadvantages of oligo probes based on single-copy sequences and oligo probes based on repetitive DNA sequences (Revised version, lines 93). Then some other revisions have been described as follows:

  1. Lines 52-58 are rewritten in the revised version, which is colour highlighted and mainly emphasizes the drawbacks of the traditional repetitive sequences and BAC probes focusing on their problems for preparation and application in plants.
  2. In line 71, one reference was added as Ref. [17], which indicates that Chorus2 is one of the most commonly used pipelines for designing oligo probes. It should be mentioned alongside other probes.
  3. In Line 80, the title has been changed to “De novo oligo probe synthesis” instead of “Oligo Probe Synthesis”. The “De novo oligo probe synthesis” title is a better fit in this section because all processes described here are new and helpful for probe design.
  4. In line 83, one reference was added as Ref. [20], which is related to describing the benefits of oligo probes, particularly their optimal length for better outcomes.
  5. Figure 1 caption is rewritten in the revised version, “Figure 1. Illustrates the oligos search, filtering and selecting potential oligos in the genome by Chorus2 pipeline” instead of “Figure 2. The flow diagram for synthesizing oligo probes using the Chorus2 platform”.Figure 2 caption is rewritten in the revised version, “Figure 2. The schematic diagram represents oligo probes library enrichment, PCR amplification for primers binding, fluorophore labelling for the de novo probes synthesis and performing FISH experiment with developed oligo probes” instead of “Figure 1. The procedure for preparing oligo probes and performing FISH technique”. These modifications are intended to clarify the content expressed in the figures more clearly.
  6. Lines 254-256 are rewritten with one reference added as Ref. [12] which introduces the importance of using oligo probes at specific concentrations for different species and cells for better FISH signals.
  7. We added four new citations of Ref. [22], [75], [76] and [82]. [22] describes the important application of ND-FISH based on oligo probes from repetitive DNA sequences. Ref. [75] and [76] have been published recently and are good examples of oligo probes application in citrus. Ref. [82] describes the oligo probes have excellent performance in constructing ancestral chromosomes. 
  8. In line 380, we merged 4.2 and 4.3 into one part and made some logical modifications (Revised version, lines 286).
  9. “Bio-dUTP” has been replaced by three fluorophore markers such as “Biotin or digoxigenin or fluorescein probe”, which is more comprehensively described in Figure 2.
  10. All small changes have been made with the colour highlighted.

We appreciate your positive feedback in the comments and suggestions section and thank you for your time and effort in reviewing our manuscript.

Best regards,

Prof. Chunli Chen

Huazhong Agricultural University and

Corresponding author of the manuscript

Round 3

Reviewer 1 Report

This revised version has been rewritten well. It can be accepted for publication.